# Fermentation Products Originated from *Bacillus subtilis* Promote Hepatic–Intestinal Health in Largemouth Bass, *Micropterus salmoides*

**DOI:** 10.3390/biology14060646

**Published:** 2025-06-02

**Authors:** Kaifang Liu, Shubin Liu, Dexiang Feng, Pengwei Xun, Hanjun Jiang, Yanwei Zhang, Gaoliang Yuan, Xusheng Guo

**Affiliations:** 1College of Fisheries, Xinyang Agriculture and Forestry University, Xinyang 464000, China; liukf@xyafu.edu.cn (K.L.); 2022210002@xyafu.edu.cn (P.X.); 2021210003@xyafu.edu.cn (H.J.); 2024210005@xyafu.edu.cn (Y.Z.); 2024210006@xyafu.edu.cn (G.Y.); 2Fishery Biological Engineering Technology Research Center of Henan Province, Xinyang 464000, China; 3Shanghai Fisheries Research Institute, Shanghai Fisheries Technical Extension Station, Shanghai 200433, China; liushubin@caas.cn

**Keywords:** antibacterial fermentation products, *Bacillus subtilis*, *Micropterus salmoide*, gut microbiota, hepatic-intestinal health

## Abstract

This study investigates the effects of fermentation products (FPs) from *Bacillus subtilis* on the antioxidant capacity and gut microbiota of Largemouth Bass (*M. salmoides*), with the fish fed experimental diets containing 0, 1%, 3% and 5% FPs. Although the short-term administration of FPs had no significant impact on the growth performance of Largemouth Bass, it reduced liver injury markers, enhanced antioxidant capacity, increased the expression of immunity and lipolysis-related genes, and significantly altered the diversity and structure of gut microbiota, with different groups having different microbial markers. The addition of FPs resulted in *Paenibacillus* being the most significant microbial marker, activated certain pathways, and changed microbial abundances in the Bugbase analysis. Our findings show that a dietary supplement of 1–3% FPs can enhance the antioxidant ability and improve the liver and intestine health of *M. salmoides*, making it a promising feed additive in aquaculture.

## 1. Introduction

The majority of diseases afflicting aquatic animals are bacterial infections. To reduce mortality rates and minimize economic losses, antibiotics have been widely adopted by farmers as the primary treatment strategy [1]. Antibiotics are valued for their cost-effectiveness, rapid action, and ease of administration, making them a common choice for combating bacterial infections [2]. However, excessive antibiotic use has led to the emergence of antibiotic-resistant bacteria, complicating disease control while causing drug residues, environmental contamination, immune suppression in aquatic animals, and risks to human health [3]. Consequently, reducing antibiotic reliance has become imperative in aquaculture, driving the search for novel antibacterial alternatives [4].

The fermentation process involves microbial growth on a solid matrix in the presence of water, followed by the separation and collection of bioactive compounds from the supernatant [5]. Fermentation products (FPs) are active substances with antimicrobial properties against exogenous pathogens, offering a promising source of antibacterial agents due to their high efficacy and low resistance potential [6,7]. Previous research has demonstrated that dietary supplementation with FP enhances growth performance in various livestock, including pigs, preweaned calves, and broilers, serving as a cost-effective strategy for disease prevention [8,9,10,11]. Current research primarily focuses on long-term FP administration (over 56 days) for prophylactic purposes [12,13]. However, prolonged use of feed additives or immune enhancers may induce immune suppression or dysregulation in hosts, counteracting their intended benefits [14]. In contrast, short-term supplementation strategies—such as dietary inclusion of *Artemisia annua* alcoholic extract in juvenile Nile tilapia (*Oreochromis niloticus*) or DL-methionine in rainbow trout—have proven effective in boosting immunity and growth without adverse effects [15,16]. Hence, optimizing the duration of additive use could improve cost-efficiency and practicality in aquaculture.

*Bacillus* species are widely employed in biological control and animal feed due to their diverse bioactive metabolites, which inhibit common pathogens and present a sustainable alternative to conventional antibiotics [17,18,19]. For instance, antimicrobial peptides from soil-derived *Bacillus amyloliquefaciens* RX7 exhibit remarkable thermal and pH stability [20]. *Bacillus* strains synthesize antibacterial compounds such as bacteriocins, iturins, surfactins, chitinases, and proteases, alongside enzymes like β-mannanase and aminopeptidase [20]. *Bacillus* strains synthesize antibacterial compounds such as bacteriocins, iturins, surfactins, chitinases, and proteases, alongside enzymes like β-mannanase and aminopeptidase [21]. Their extracellular metabolites also modulate gut microbiota, influencing host health [22]. While antibiotics disrupt the gut microbial equilibrium and persist in aquatic products, FPs promote beneficial bacterial proliferation, restoring microecological balance and enhancing animal growth [23,24].

The Largemouth Bass (*Micropterus salmoides*) is a major freshwater aquaculture species in China, with an annual yield exceeding 800,000 tons, ranking as the highest-producing carnivorous freshwater fish [25]. However, rapid industry expansion has intensified disease outbreaks, including *Micropterus salmoides* rhabdovirus (MSRV) and scuticociliatosis, posing significant challenges due to the lack of effective vaccines and reliance on antibiotics [26,27,28]. This underscores the urgent need for natural alternatives to synthetic drugs. This study investigates the effects of *B. subtilis*-derived FPs on the growth, and hepatic and intestinal health of *M. salmoides* and explores how their use correlates with gut microbiota modulation. The findings aim to advance the application of FP as a sustainable feed additive in aquaculture for holistic animal health improvement.

## 2. Materials and Methods

### 2.1. Ethical Approval

All animal procedures in this study were conducted according to the animal husbandry guidelines of Xinyang Agriculture and Forestry University, Xinyang City, Henan Province, China. The studies in animals were reviewed and approved by the Laboratory Animal Welfare and Ethical Reviewing Committee of Xinyang Agriculture and Forestry University (No. 2024-AF-XYAFU-001).

### 2.2. Antibacterial Fermentation Products Information

The fermentation product (FP-WeiGuangSu, purity ≧ 99.9%) was obtained from Jiangsu Jiurun Biotechnology Co., Ltd. (Jiangsu, China), mainly through isolation from the *Bacillus subtilis* strain (the strain shared approximately 99.86% with *Bacillus subtilis* DSM 10, *Bacillus subtilis* NBRC 13719 and *Bacillus subtilis* JCM 1465), which originated from a soil environment and is maintained in our laboratory. We selected *Aeromonas hydrophila* (GenBank: GU563992.1), *Aeromonas salmonicida* (GenBank: MK034473.1) and *Aeromonas veronii* (GenBank: OQ283658.1), which are pathogenic to Largemouth Bass and are maintained in our laboratory. According to the pre-experiment, the concentration of the pathogenic bacteria was adjusted to 10^6^ CFU/mL. The vitality of bacteria in the supplemented diet was evaluated by plate counting on LB medium agar. We prepared ddH_2_O with pH values ranging from 3 to 11 using 1 mol/L HCl and 5 mol/L NaOH, respectively, and dissolved the FP in the ddH_2_O of different pH values. After dissolving FP in ddH_2_O, it was incubated in water baths at 20 °C, 40 °C, 60 °C, 80 °C, and 100 °C, respectively, for 1 h. We then dissolved the FP in solutions of NaCl, KCl, MgCl_2_, and CaCl_2_ with concentrations of 50 mmol/L, 100 mmol/L, 150 mmol/L, and 200 mmol/L, respectively. We then added 10 μL of FP with a concentration of 1 mg/mL (treated as above) to the wells of LB solid plates containing the three pathogenic bacteria, and detected the effect of the FP on acid by measuring the diameters of the inhibition zones. The diameter of the inhibition zone was calculated by vernier calipers after culturing the plates for 24 h under 30 °C [29].

### 2.3. Experimental Diets and Animals

The juvenile *M. salmoides* laboratory facility of the Xinyang Agriculture and Forestry University (Henan, China) was used to conduct the experiment. Healthy, uniformly sized *M. salmoides* (24.54 ± 0.06 g/fish) were divided into four groups (Control, H1, H2 and H3) randomly and fed twice daily to satiation with four experimental diets. The control diet was a Largemouth Bass commercial diet (containing 39.64% crude protein, 12.71% crude lipid, 14.01% ash and 10.08% moisture, from Guangdong Haid Group Co., Guangzhou, China), while the H1, H2 and H3 diets were the commercial diet with 1%, 3% and 5% FP. Each group of fish consisted of three repetition tanks, and each tank contained 15 fish. The experimental period was 15 days. The following conditions were maintained during the trial period: water temperature, 26–28 °C; pH 6.8–7.2; 14 h/10 h light/dark cycle; dissolved oxygen more than 7.0 mg/L; and ammonia nitrogen content less than 0.15 mg/L.

### 2.4. Growth Performance Measurements and Sampling

Before the experiment began, six fish were randomly selected. The intestinal contents were taken 4 to 6 h after full-feeding and used for microbiome analysis; these fish were named the C0 group. At the end of the feeding experiment, we measured the body weight and calculated the weight gain (WG), specific growth rate (SGR) and feed conversion ratio (FCR) of *M. salmoides* using the following formulas:$$WG(\%)=\frac{final\;body\;weight-initial\;body\;weight}{initial\;body\;weight}\times 100\%$$$$SGR(\%/day)=\frac{Ln(final\;body\;weight)-Ln(initial\;body\;weight)}{experimental\;period}\times 100\%$$$$FCR=\frac{total\;feed\;weight}{final\;body\;weight-initial\;body\;weight}$$

After that, we randomly selected half of the fish in each tank for routine sampling, while the other fish were fed to satiety before collecting the intestinal contents for microbiome detection 4 h later. All samples were quickly frozen directly in liquid nitrogen and finally transferred to −80 °C. Routine sampling was mainly used for serological tests and measuring liver antioxidant capacity and immune-related gene expression. Each fish was subjected to tail vein blood collection after anesthesia, and the collected blood was placed in a 1.5 mL centrifuge tube and left to stand overnight in 4 replications. After centrifugation at 2500 rpm for 15 min, the supernatant was taken for serological index detection. Subsequently, the same portion of the liver was evenly divided into two parts and quickly frozen in liquid nitrogen; of these, one part was used for antioxidant capacity detection, and the other was used for immune-related gene expression detection.

### 2.5. Serological Indicator

The commercial kits (Jiancheng Biotech Co., Nanjing, China) were employed to measure the activities of serum alanine aminotransferase (ALT), aspartate aminotrans ferase (AST), albumin (ALB), alkaline phosphatase (AKP), and antioxidant-related enzyme activity, such as superoxide dismutase (SOD), catalase (CAT) and glutathion peroxidase (GSH-px), as well as total antioxidant capacity (T-AOC).

### 2.6. Hepatic Antioxidant Indices

In order to appraise the hepatic antioxidant, the liver samples were subjected to ultrasonic homogenization by adding sterile phosphate-buffered saline (PBS) at a ratio of 1:9; the supernatant was obtained by centrifugation at 12,000 rpm for 10 min at 4 °C for detection. Then, the concentration of malondialdehyde (MDA), the activities of superoxide dismutase (SOD), catalase (CAT) and glutathion peroxidase (GSH-px), as well as total antioxidant capacity (T-AOC), were measured by using commercial assay kits (Jiancheng Biotech Co., Nanjing, China) according to the manufacturer’s instructions.

### 2.7. RNA Extraction and Quantitative PCR Analysis

For the analysis of the expression of immune-related genes and lipolysis-related genes, specific primers of *M. salmoides* were designed by NCBI Primer BLAST (https://www.ncbi.nlm.nih.gov/tools/primer-blast/, accessed on 18 February 2025) (Appendix A). Liver tissues were used to extract the total RNA, using RNAiso Plus (Vazyme, Nanjing, China). The quality of total RNA was measured by 2.0% agarose gel cataphoresis and microspectrophotometer (Thermo Fisher Scientific Inc., Waltham, MA, USA), and 1.20 μg of total RNA was reverse-transcribed for quantitative real-time PCR. The protocol settings for qPCR were conducted as previously described [29]. Melting curves analysis was performed to investigate amplicon specificity and reaction product. The messenger RNA expression levels of all genes were standardized by the *β-actin* gene. Gene expression analysis was performed by the 2^−ΔΔCT^ method [29].

### 2.8. Gut Microbiota Analysis

We used 16S rRNA gene sequencing to analyze gut microbiota. Intestinal content samples were collected at 4–6 h post the last feeding. Samples from five fish per tank were pooled. Intestinal bacterial DNA was extracted by using TIANamp Bacteria DNA Kit (TianGen, Beijing, China), according to the manufacturer’s instructions. The V3-V4 region of 16S rRNA was amplified by PCR using the primers 338 F (ACTCCTACGGGAGGCAGCAG) and 806 R (GGACTACHVGGGTWTCTAAT) and sequenced in the Illumina Miseq platform. The protocol settings for qPCR were as previously described [29]. Then, the raw pair-end reads were subjected to a quality-control procedure using UPARSE and assembled into operational taxonomic unit (OTUs). The qualified reads were clustered to generate OTUs at the 97% similarity level using UPARSE. A representative sequence of each OTU was assigned to a taxonomic level in the Ribosomal Database Project (RDP) database using the RDP classifier. Principal coordinate analysis (PCoA) and non-metric multidimensional scaling analysis (NMDS) were performed at the OTU level using R language. Shannon and Simpson diversity indices were calculated using Qiime software (QIIME 2) to evaluate Alpha diversity. Species classification analysis of OTUs was conducted using the RDP classifier to obtain taxonomic information and to statistically analyze the community composition at the phylum and genus levels, respectively. LEfSe (LDA Effect Size) was used to analyze changes in the core microbiota among all groups. PICRUSt2 was used to predict the functions of microorganisms based on OTU data.

### 2.9. Statistical Analysis

The results are presented as the means ± SEM of the mean. Statistical analysis was carried out by SPSS 22.0 (IBM, USA). All experimental data were checked for variance homogeneity and normal distribution [29]. Differences among groups were evaluated using one-way analysis of variance (ANOVA). Tukey’s HSD test was conducted to assess the significance at a 5% level of confidence. In order to explore the relationships between the core gut microbiota and serological indicators. Pearson correlation coefficients were calculated after the removal of outlier values by Grubbs’ test (*p* < 0.05).

## 3. Results

### 3.1. FP Tolerance Test and Growth Performance

The effects of different temperatures, pH and cations on the antibacterial effect of the FP after 24 h of culture are shown in Appendix A. Except that the antibacterial effect on the three pathogenic bacteria was significantly weakened at a high temperature of 100 °C (*p* < 0.05), the other conditions had no significant effect on the antibacterial effect of the FP.

The addition of the FP to the feed of *M. salmoides* did not significantly change the weight gain rate, specific growth rate and feed conversion rate in the short term (*p* > 0.05). However, the weight gain rate of the 1% FP group (H1) was the highest, with a value of 207.35 ± 15.86% (Table 1).

### 3.2. Serological Indicators

The serological indicators representing liver injury showed that the contents of ALT, AST and AKP in the groups with added FP were lower than those in the control group (Figure 1). Among them, the H1 group was significantly lower than the control group (*p* < 0.05). There was no significant difference in the content of ALB among the groups (*p* > 0.05). The serological indicators showed that the antioxidant capacity (SOD, CAT and T-AOC) of the H1 group was significantly higher than that of the control group, while the values for the H2 and H3 groups were only significantly higher than the control group for the contents of CAT and T-AOC, respectively (*p* < 0.05).

### 3.3. Hepatic Antioxidant Capacity

The results for the detection of the antioxidant capacity of liver tissue showed that the addition of FP could reduce the MDA content of the liver (Figure 2A) and significantly increase the contents of SOD, CAT and GSH-px, while the T-AOC content was higher than that of the control group, but there was no significant difference (Figure 2B–E, *p* < 0.05).

### 3.4. Gene Expression

The gene expression results for immune-related genes indicated that the addition of FP in H1 and H2 groups could increase the expression levels of *il-10* and *IgM* in the liver to varying degrees (Figure 3A). Among them, the expression level of *il-10* in group H2 was significantly higher than that in the control group, while the expression level of *IgM* in group H1 was significantly higher than that in the control group. The results of lipid metabolism-related genes showed that the addition of FP had no significant effect on fat synthesis, but affected the expression of genes related to fat decomposition, such as *pparα* and *cpt* (Figure 3B). Compared with the control group, groups H1 and H3 had significantly increased expression levels of *pparα*, and group H1 also had significantly increased expressions level of the *cpt* gene.

### 3.5. Gut Microbiota Diversity and Distribution

The results of gut microbiota diversity showed that there was no significant difference in alpha diversity (ACE, Chao1, Simpson, and Shannon) in the control group compared to the initial state (C0 group), while the alpha diversity in the group with added FP was increased compared to the control group (Table 2, *p* < 0.05). In terms of beta diversity, the control group was significantly separated compared to the C0 group, while the group with added FP was clustered and distant from the control group and the C0 group (Figure 4). The results of microbiota species distribution showed that, at the phylum level, the control group had significantly reduced abundance of Proteobacteria and increased abundance of Fusobacteria compared to the C0 group (Figure 5A). Compared to the control group, the group with added FP had significantly increased abundance of Firmicutes (*p* < 0.05). At the genus level, the control group had significantly reduced abundance of *Plesiomonas* compared to the C0 group (Figure 5B, *p* < 0.05). The microbiota abundances that significantly increased in the group with added FP included *Mycoplasma*, *Enterococcus*, *Paenibacillus*, *Acinetobacter*, and *Bacillus*. Among them, *Cetobacterium* was associated with the control group and the C0 group, and *Plesiomonas* was associated with the C0 group (Figure 5C).

### 3.6. Bio-Markers and Predicted Functions

The LEfSe analysis results indicated that when the threshold was 4, the microbial biomarker of the C0 group was *Plesiomonas*, that of the control group was *Cetobacterium*, and those of the H1 group were *Bacillus* and *Mycoplasma*. The microbial biomarkers of the H2 group were *Acinetobacter*, *Paenibacillus* and g_unclassified_*Rhizobiaceae*, and that of the H3 group was *Enterococcus* (Figure 6A,B). The random forest results showed that the most dominant microbial biomarkers after the addition of FP were *Paenibacillus*, *Mycoplasma*, *Enterococcus* and *Rheinheimera* (Figure 6C).

The results of microbiota function prediction showed that at the third level, compared with the control group, in the group with added FP, biosynthesis of secondary metabolites, biosynthesis of antibiotics, biosynthesis of amino acids, and the two-component system were the main activated KEGG metabolic pathways, while pyrimidine metabolism and quorum sensing were inhibited (Figure 7A). The Bugbase analysis results showed that, compared with the control group, the abundance of anaerobic bacteria decreased in the FP group, while the microbial abundances of mobile-element-containing and oxidative-stress-tolerant phenotypes increased (Figure 7B). The differences in KEGG pathways of microorganisms of the H1 and H2 groups relative to the control group were analyzed separately. The results showed that the pathways of biosynthesis of secondary metabolites, biosynthesis of antibiotics, and biosynthesis of amino acids were significantly activated (Figure 7C,D).

The results of the Pearson’s correlation are summarized in Figure 8. The *Mycoplasma* had a significantly positive correlation with ALB (0.970) and SOD (0.979), but was negatively correlated with ALT (−0.985), while *Cetobacterium* had significantly negative correlations with ALB (−0.961) and T-AOC (−0.976). An extremely significant positive correlation was found in CAT between *Bacillus* (0.969) and *Paenibacillus* (0.962), but strongly negative correlations were found in AKP (−0.986 and −0.954).

## 4. Discussion

As feed additives, fermentation products must possess good thermal stability, acid-base tolerance and other characteristics. Bradley classified microbial antibacterial fermentation products into two categories based on their molecular weights: low-molecular-weight chemical compounds, and high-molecular-weight proteins [30]. Generally, high-molecular-weight proteins are sensitive to trypsin, but have good thermal stability. The stability (antibacterial effectiveness) of high-molecular-weight proteins varies with their purity, environmental pH, ionic strength, and the presence or absence of protective factors; it also determines their potential and application scope in practical applications [31]. Nevertheless, the differences among FPs from different sources are still obvious. The diameter of the inhibition zone of the antimicrobial substance temporin-GHb from the skin of the Hainan Marsh Frog decreased from 9 mm to 5 mm after treatment at 80 °C for 20 min [32]. The antimicrobial substance produced by *Bacillus licheniformis* has strong stability, and external factors such as high temperature (100 °C), acid-base changes, etc., cannot significantly reduce its antibacterial titer [33]. The antimicrobial substance Lc-NKlysin-1a obtained from the Large Yellow Croaker (*Larimichthys crocea*) has good tolerance to temperature, pH and salt [34]. The antibacterial activity of the bovine erythrocyte hemoglobin-derived antimicrobial substance did not change significantly after treatment at high temperatures of 75–100 °C, at high concentrations of NaCl of 80–100 mmol/L, or at pH 4.0 or pH 10.0 [35]. Dashper et al. [36] found that the antibacterial activity of the antimicrobial substance kappacin was only active in an acidic environment, suggesting that pH affects the binding of the antimicrobial substance to cell membranes. Charge changes caused by ions can reduce antibacterial activity. Most β-defensins lose their biological activity at a sodium chloride concentration of 150 mmol/L [37]. In this study, the FP isolated from *Bacillus subtilis* still had strong antibacterial activity after treatment at 80 °C, treatments with strong acids and bases, and treatments with 150 mmol/L cations (Na^+^, Mg^2+^, Ca^2+^ and K^+^). These tolerance test data prove, to a certain extent, that the FP produced by *B. subtilis* has better thermal stability, acid-base tolerance and various cation tolerances, making it suitable as a feed additive.

The levels of alkaline phosphatase (AKP), alanine aminotransferase (ALT), and aspartate aminotransferase (AST) in serum can be used as indicators of liver damage. Elevated levels of these three enzymes may suggest liver degeneration and/or damage and can be used to assess the toxic effects of feed additives on the liver [38]. In this study, all three addition doses of FP showed reduced liver damage indicators, indicating that the FP as a feed additive has no liver toxicity. Albumin is a globulin with antibody activity or a chemical structure similar to antibodies. It is widely present in the serum of vertebrates and plays an important role in the body’s immune system. Its content can be used as a marker to evaluate the immune function of animal serum [39]. However, the addition of FP in this study did not cause a significant change in this immune indicator in the serum, possibly because the feeding time was too short to cause a significant effect [40]. Previous studies have shown that reactive oxygen species (ROS) produced as metabolic by-products can oxidize cellular components such as lipids, proteins, and DNA. When their production rate exceeds the removal rate, they pose a threat to the structure and function of cells [41]. SOD is an important component of the antioxidant enzyme system in animals, and catalyzes the disproportionation of superoxide anion radicals to generate oxygen and hydrogen peroxide [42], and GSH-Px is an important peroxide-decomposing enzyme that can protect the structure and function of cell membranes [43]. As one of the final products of lipid peroxidation, the MDA content reflects the effect of oxidative damage [44]. In this study, regardless of whether it was in serum or liver tissue, different addition concentrations of FP were able to increase the levels of antioxidant-related indicators (SOD, CAT, GSH-px, and T-AOC), and the content of MDA in liver tissue was significantly reduced, indicating that FP can improve the antioxidant capacity of fish. Previous studies have reported that 90 mg/kg of antimicrobial substance APSH-07 can promote the growth of Large Yellow Croaker and improve antioxidant and stress resistance abilities [45]. Chen et al. [46] fed tilapia two FPs, Natucin C and Natucin P, and found that their mixture can significantly improve the antioxidant capacity and innate immunity of tilapia, concluding that they can be used as a potential alternative to antibiotics. Although the main function of antimicrobial substances is to inhibit and kill pathogenic bacteria and prevent viruses from binding to the host [47], many current studies have shown that as feed additives, they can improve the antioxidant capacity and immunity of the host. Many studies have shown that antioxidant capacity is often closely related to immunity and lipid metabolism [48]. At present, this study also shows that the addition of FP can increase the expression level of immune-related genes in Largemouth Bass and can promote fat decomposition. As reviewed by Wang et al. [49], antimicrobial substances as a feed additive can improve the immunity of farmed animals and reduced fat accumulation. However, the main mechanisms of these are still unclear and may be related to the gut microbiota of the host. Meanwhile, in this study, Pearson correlation coefficients showed that there was a significant correlation between the core gut microbiota of Largemouth Bass fed FP and serological related indicators.

The gut microbiota is regarded as the “second genome” and is closely related to the host’s health. The abundance of core gut microbiota species fluctuates within a small range at different time stages. Through interactions with the host’s digestive system and immune system, etc., it establishes a balanced relationship with the host’s health [50]. When the balance of the human gut microbiota is disrupted, the species and abundance of the gut microbiota change significantly, thereby causing the occurrence of inflammation, aging, obesity, immune disorders, and metabolic abnormalities [51]. After exogenous FPs enter the digestive tract, they are cleaved by digestive enzymes into antimicrobial substance with smaller molecular weights. While microorganisms produce antimicrobial substance by acting on proteins, antimicrobial substances can also exert an influence on the structure of the gut microbiota [52]. The metabolites of the gut microbiota are closely related to the pathogenesis and treatment of various diseases, and some FPs may exert therapeutic effects by influencing the metabolites of the gut microbiota [53]. In this study, the FP affected the biosynthesis of secondary metabolites in the metabolic pathways of the gut microbiota. These results are consistent with studies in mammals, indicating that the FP affected the metabolites of the gut microbiota. In addition, in previous studies, FPs destroyed the structure of harmful bacteria, eliminated reactive oxygen species in the intestinal tract, and were decomposed and utilized as substrates for specific microorganisms, thereby regulating the gut microbiota [44]. They also regulate the abundance and diversity of the gut microbiota through different mechanisms, including as antimicrobial agents, antioxidants, and as a source of nutrition for beneficial microbiota [54]. However, in the existing studies applying FPs in aquatic farmed animals, reports on the gut microbiota are limited.

Previous studies have reported the sequencing and characterization of hepcidin from the liver of *Acrossocheilus fasciatus*, and it was found that hepcidin recovered the diversity and composition of gut microbiota [55]. The addition of 90 mg/kg of antimicrobial substance APSH-07 in the diet changed the phylogenetic diversity of the gut microbiota of *L. crocea* [45]. An antimicrobial substance improved the structure and composition of the gut microbiota, increasing the microbial community diversity of *Cherax quadricarinatus* [56]. The addition of dietary FPs (antimicrobial peptides, a 6 kDa and 5 kDa compound from the intestine) increased the abundance of Acidobacteria, Proteobacteria, and Patescibacteria in the gut microbiota of *Ctenopharyngodon idelus* and decreased the abundance of Fusobacteria and Firmicutes [57]. The polypeptide S100 significantly upregulated the abundance of Cyanobacteria in the gut microbiota of *Penaeus vannamei*, and a 1% dose could significantly increase Pseudomonadaceae and Xanthomonadaceae, while reducing Vibrionaceae [58]. In this study, FPs as feed additives changed the structure and species distribution of the gut microbiota of *Micropterus salmoides*, and there were obvious gradient changes. The core gut microbiota of each addition dose had unique biomarkers. At the same time, it was found in this study that the structure and distribution of the core gut microbiota changed by the FPs were closely related to the metabolic pathways for the biosynthesis of antibiotics and amino acids. This might be because the amino acid residues on the protein or chemical compounds of the FP combined with substances such as sugars and lipids to form derivatives. Due to these derivatives having some special structures, they also play a certain regulatory role on the gut microbiota [59]. Thus, it can be seen that FPs can regulate the structure of the gut microbiota by inhibiting the reproduction of harmful bacteria and promoting the growth of beneficial bacteria in the intestine, thereby maintaining the homeostasis of the intestinal environment. However, in the current studies, the specific mechanism by which FPs regulate the richness and diversity of the gut microbiota has not been completely and specifically elaborated, and the specific selection mechanism for beneficial and harmful bacteria still needs further research.

## 5. Conclusions

The short-term administration of FP-WeiGuangSu resulted in decreased liver injury markers and MDA content, enhanced antioxidant capacity and immune-related gene expression, and significantly changed the diversity and structure of gut microbiota, with *Paenibacillus* as the most significant microbial marker. It activated related pathways, and decreased anaerobic bacteria while increasing specific phenotypes of microorganisms. These findings clarified that a dietary supplementation of 1–3% FP could be utilized to boost the antioxidant capacity, along with the liver and intestine health of *M. salmoides* in the aquaculture field, and this FP might be considered as a promising feed additive or immunopotentiator in aquaculture. Through the evaluation methods of this study, we found that the indicators of intestinal microbiota demonstrated the effects of FPs on Largemouth Bass more sensitively and accurately.

## Figures and Tables

**Figure 1 biology-14-00646-f001:**
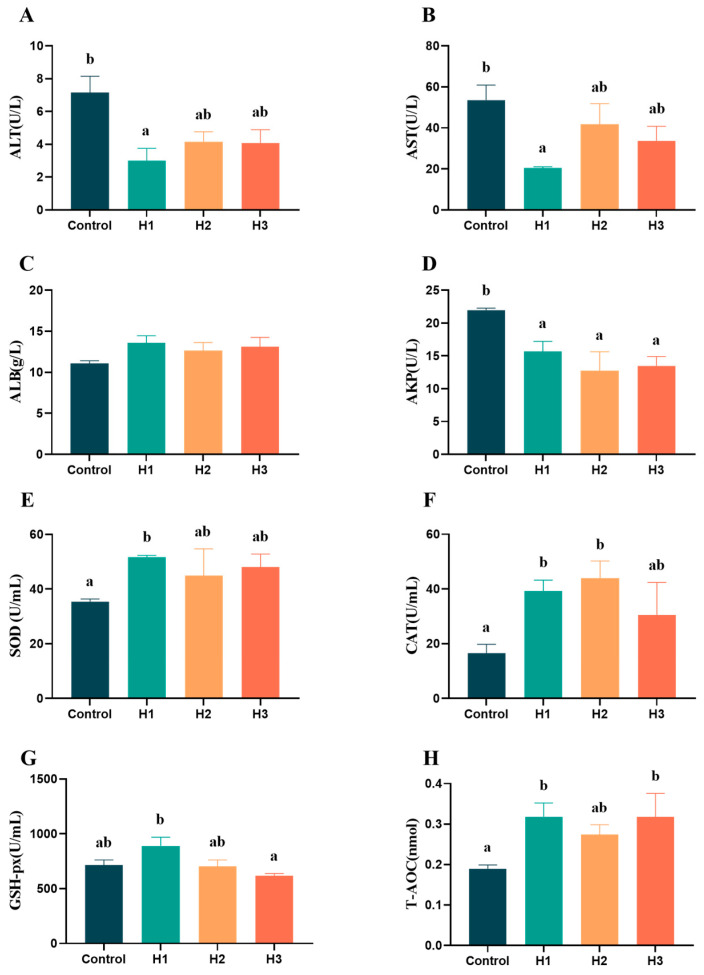
Effects of 0% (control), 1% (H1), 3% (H2) and 5% (H3) FP diets on serological indicators of *Micropterus salmoides*. The content or activity of (**A**) alanine aminotransferase, (**B**) aspartate aminotransferase, (**C**) albumin, (**D**) alkaline phosphatase, (**E**) superoxide dismutase, (**F**) catalase, (**G**) glutathione peroxidase, and (**H**) total antioxidant capacity. Data are presented as means ± SEM, and the significant differences in the results are represented by different lowercase letters (*p* < 0.05).

**Figure 2 biology-14-00646-f002:**
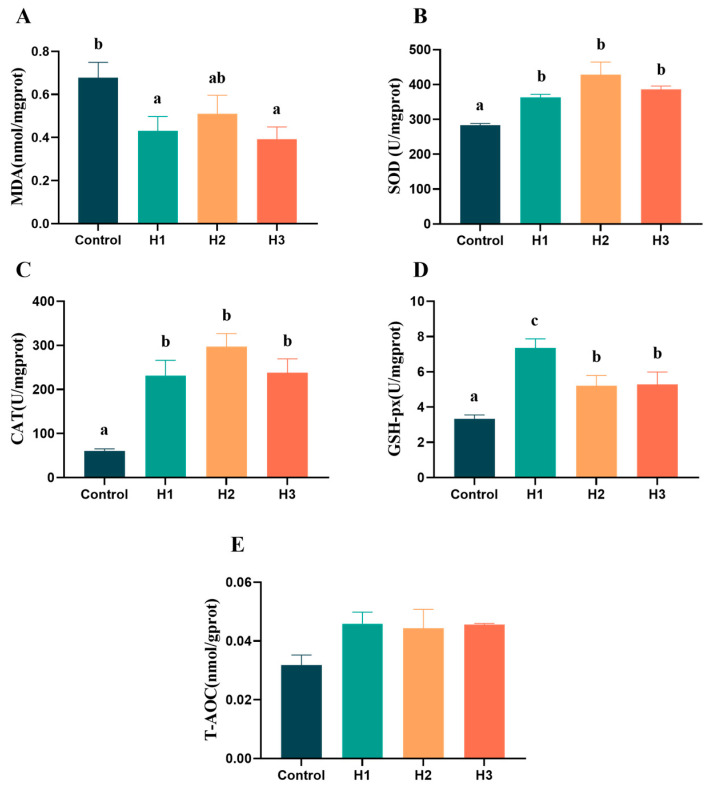
Effects of 0% (control), 1% (H1), 3% (H2) and 5% (H3) FP diets on antioxidant capacity of the liver. The content of (**A**) malondialdehyde, and the activity of (**B**) superoxide dismutase, (**C**) catalase, (**D**) glutathione peroxidase, and (**E**) total antioxidant capacity. Data represent the means ± SEM. Results with significant differences are marked with different letters (*p* < 0.05).

**Figure 3 biology-14-00646-f003:**
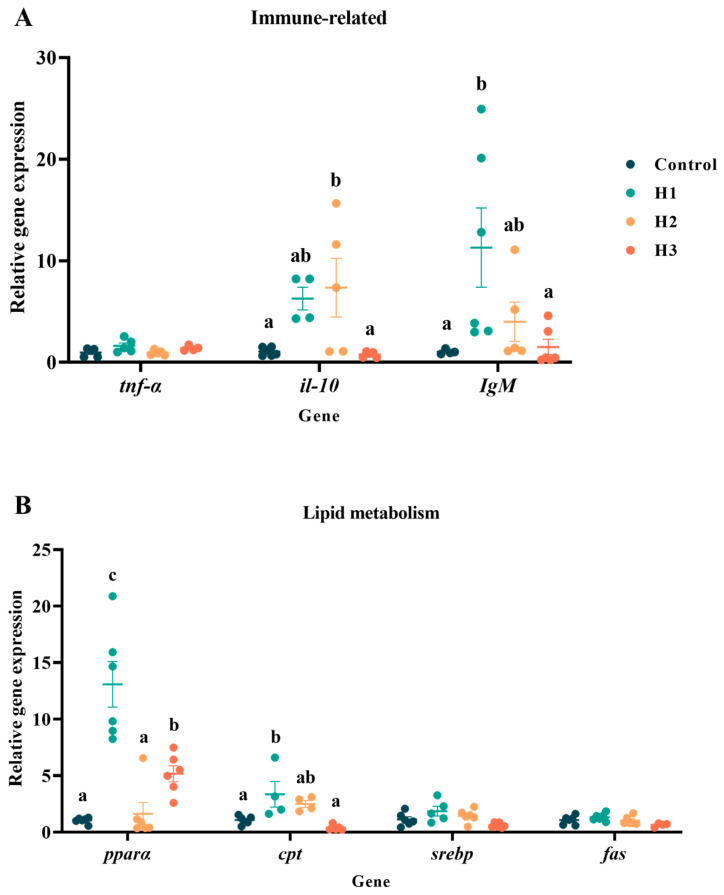
Effects of 0% (control), 1% (H1), 3% (H2) and 5% (H3) FP diets on the expression levels of (**A**) immune-related genes and (**B**) lipid metabolism-related genes of the liver. Data represent the means ± SEM. Results with significant differences are marked with different letters (*p* < 0.05).

**Figure 4 biology-14-00646-f004:**
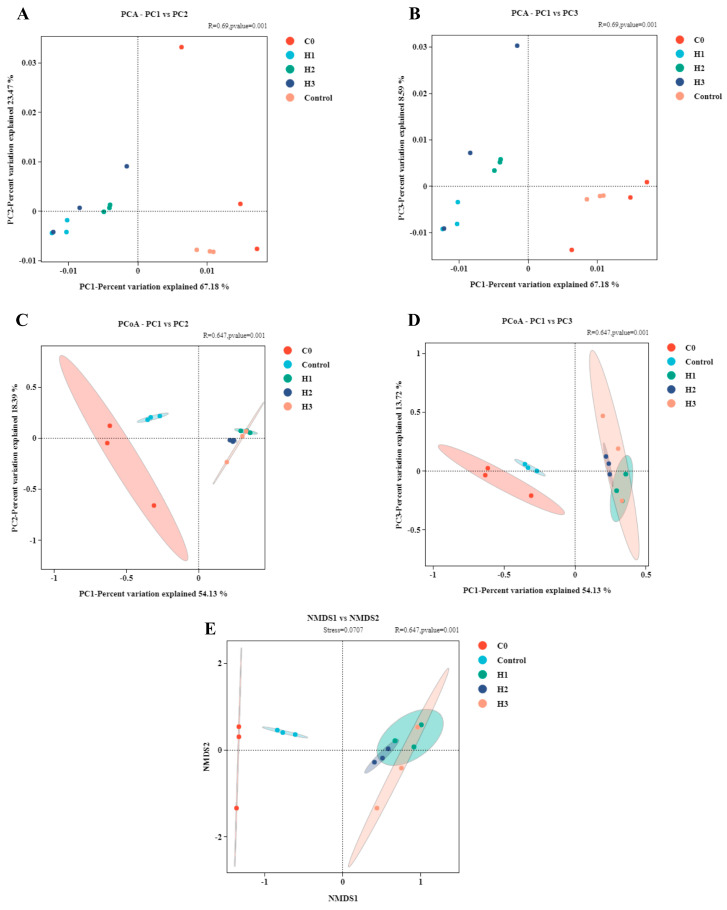
Beta diversity analysis of features based on the Bray–Curtis algorithm. Principal component analysis (PCA) based on (**A**) PC1 vs. PC2 and (**B**) PC1 vs. PC3; principal coordinates (PCoA) analysis using genus level in (**C**) PC1 vs. PC2, and (**D**) PC1 vs. PC3; and (**E**) non-metric multidimensional scaling (NMDS) analysis of intestinal bacterial communities of initial fish (C0 group) and experimental fish (Control, H1, H2 and H3 groups).

**Figure 5 biology-14-00646-f005:**
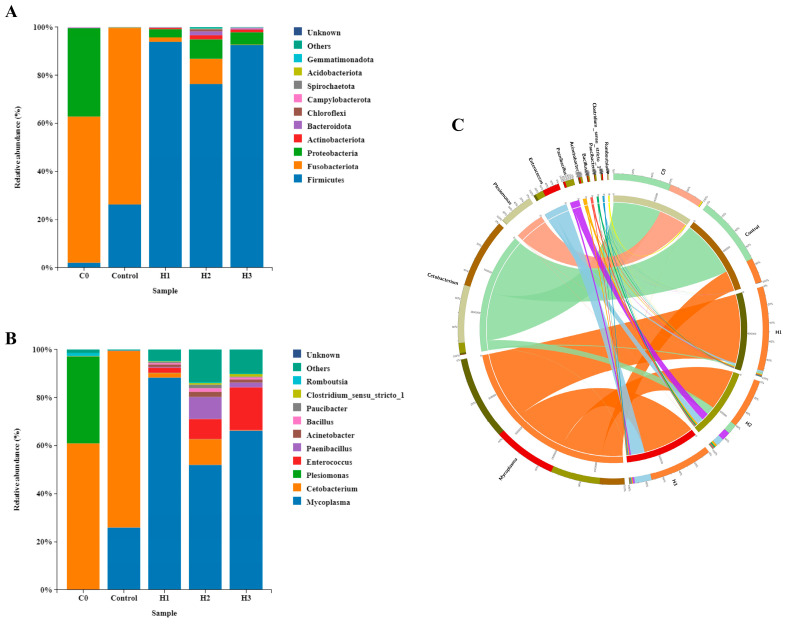
Relative abundance ((**A**) phylum, and (**B**) genus) of the dominant bacterial communities (top 10) in the intestine, and (**C**) circos plot of the top 10 abundant bacterial features at the genus levels in initial fish (C0 group) and experimental fish (Control, H1, H2 and H3 groups).

**Figure 6 biology-14-00646-f006:**
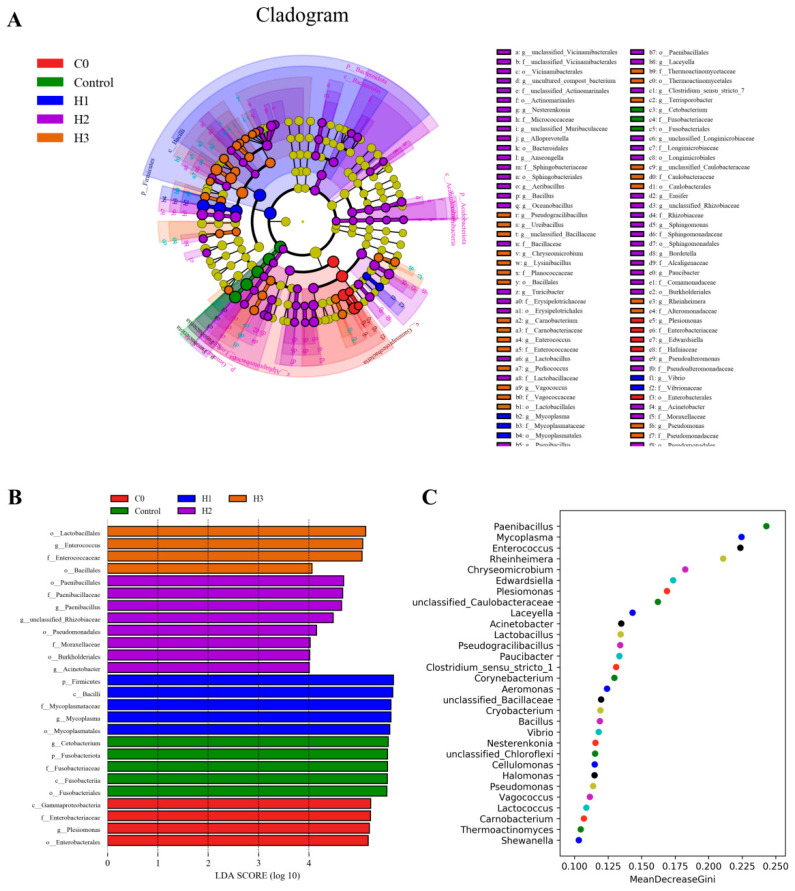
Biomarker analysis among initial fish (C0 group) and experimental fish (Control, H1, H2 and H3 groups). (**A**) Cladogram of LEfSe using an LDA score threshold of > 2. (**B**) Bacterial taxa differentially displayed in the intestines were identified by LEfSe using an LDA score threshold of >4. (**C**) Random Forest analysis showing the importance ranking of biomarker species.

**Figure 7 biology-14-00646-f007:**
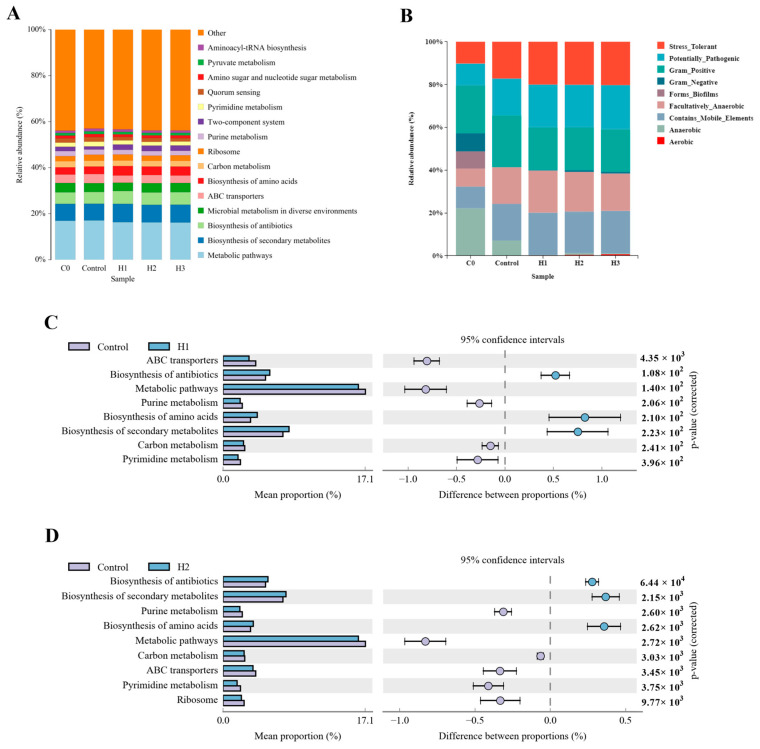
(**A**) KEGG analysis at level 3, and (**B**) Bugbase phenotype prediction among initial fish (C0 group) and experimental fish (Control, H1, H2 and H3 groups). (**C**,**D**) Shown the left are the abundance ratios of different functions in two groups (H1 and H2) compared to the control. The middle figure represents the difference ratio of functional abundance within a 95% confidence interval. The *p*-value is shown on the right.

**Figure 8 biology-14-00646-f008:**
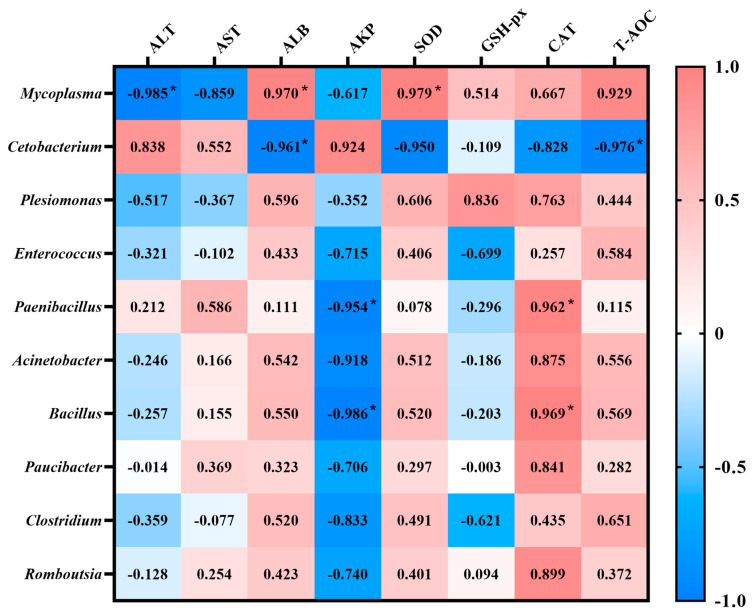
Pearson correlation coefficients between the intestinal core microbiota and serological indicators in groups fed with different doses of FP. Significant coefficients: *, *p* < 0.05.

**Table 1 biology-14-00646-t001:** The effects of different doses of FP on the growth of *Micropterus salmoides*. Data represent the means ± SEM (12 fish). Results with significant differences are marked with different letters (*p* < 0.05).

Parameters	Control	H1	H2	H3	F	*p*
Weight gain (%)	167.81 ± 5.89	207.35 ± 15.86	176.96 ± 10.97	173.19 ± 22.99	0.520	0.508
Specific growth rate (%/day)	0.07 ± 0.00	0.08 ± 0.00	0.07 ± 0.00	0.07 ± 0.01	0.510	0.503
Feed conversion rate	1.06 ± 0.01	1.07 ± 0.04	1.05 ± 0.04	1.04 ± 0.01	0.848	0.469

**Table 2 biology-14-00646-t002:** Alpha diversity indices of *Micropterus salmoides* gut microbiota fed with different diets and initial fish.

Parameters	C0	Control	H1	H2	H3	F	*p*
ACE	53.44 ± 7.34 ^a^	58.51 ± 12.41 ^a^	118.09 ± 9.38 ^b^	221.36 ± 8.38 ^c^	153.28 ± 14.59 ^b^	0.971	0.001
Chao1	53.33 ± 7.31 ^a^	62.90 ± 13.22 ^a^	118.46 ± 8.86 ^b^	221.33 ± 8.37 ^c^	153.50 ± 14.91 ^b^	0.430	0.001
Simpson	0.24 ± 0.08 ^a^	0.39 ± 0.02 ^ab^	0.23 ± 0.08 ^a^	0.70 ± 0.02 ^b^	0.46 ± 0.19 ^ab^	0.231	0.036
Shannon	0.72 ± 0.19 ^a^	0.90 ± 0.05 ^a^	1.10 ± 0.35 ^a^	3.16 ± 0.17 ^b^	2.01 ± 0.84 ^ab^	0.194	0.012

Values are expressed as the mean ± SEM. Shannon, Shannon diversity index; Simpson, Simpson’s diversity index; ACE, ACE index; Chao, Chao index. Means with different letters are significantly different at *p* < 0.05.

## Data Availability

The original contributions presented in this study are included in the article. Further inquiries can be directed to the corresponding authors.

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
