# Peer review of "Fermentation Products Originated from Bacillus subtilis Promote Hepatic–Intestinal Health in Largemouth Bass, Micropterus salmoides"

_biology, 2025, doi:10.3390/biology14060646_

Round 1
Reviewer 1 Report (Previous Reviewer 1)
Comments and Suggestions for Authors
I have reviewed the updated version of the paper and the authors' response. But unfortunately, I have to repeat that I stand by my opinion and the authors' response did not convince me. I have to repeat it again.:
1) Despite the fact that the research was carried out qualitatively using modern methods, it is not clear what it is about. Bacillus subtilis bacteria can produce various antimicrobial peptides: subtilin, subtilosin A, ericin, mercacidin, subtilomycin, sublancin, epidermin, and others. There may also be those that have not been described before.
All of them have their advantages and disadvantages. However, the results of this work cannot expand our knowledge of these peptides in any way, since it does not specify which or which peptides we are talking about.
The trade secret argument doesn't convince me in any way. We are not talking about a patent, but about scientific work in the open access: if the authors cannot disclose the composition of the product, then it must be presented on the market and its trade name must be indicated in the article in order to ensure the principle of reproducibility of the results of scientific work.
In the text of the article (lines 454-480), the authors refer to similar works that contain information about substances, for example, in the article doi: 10.1016/j.fsi.2019.06.033 the text states: «In this research, the effects of a commercial polypeptide (Polypeptide S100) whose main components are AMPs on the growth, antibacterial immune and intestinal microbial of Litopenaeus vannamei were study».
2) The main changes made by the authors to the manuscript are the replacement of the abbreviation "Antibacterial Fermentation Products" (AFP) with "Fermentation Products" (FP) throughout the text. As well as a not very successful grammatical construction in the introduction.
3) Based on the phrase from the Authors reply to Reviewer1: «Readers can screen antimicrobial peptides through the evaluation system established in this study.», I suggest changing the concept of the article from a research-based one to a methodological one. The authors could emphasize that they propose a methodology for assessing the suitability of peptide supplements for fish diets. The result of this work would not be a conclusion that these peptides are good and can be fed to fish, but rather an assessment of how best to evaluate fish health and compare the effectiveness of different health assessment methods. In this case, the focus would shift from the substances themselves to the methods used to assess them.
Author Response
Reviewer 1
Comments and Suggestions for Authors
I have reviewed the updated version of the paper and the authors' response. But unfortunately, I have to repeat that I stand by my opinion and the authors' response did not convince me. I have to repeat it again.:
Comments 1: 1) Despite the fact that the research was carried out qualitatively using modern methods, it is not clear what it is about. Bacillus subtilis bacteria can produce various antimicrobial peptides: subtilin, subtilosin A, ericin, mercacidin, subtilomycin, sublancin, epidermin, and others. There may also be those that have not been described before.
All of them have their advantages and disadvantages. However, the results of this work cannot expand our knowledge of these peptides in any way, since it does not specify which or which peptides we are talking about.
The trade secret argument doesn't convince me in any way. We are not talking about a patent, but about scientific work in the open access: if the authors cannot disclose the composition of the product, then it must be presented on the market and its trade name must be indicated in the article in order to ensure the principle of reproducibility of the results of scientific work.
In the text of the article (lines 454-480), the authors refer to similar works that contain information about substances, for example, in the article doi: 10.1016/j.fsi.2019.06.033 the text states: «In this research, the effects of a commercial polypeptide (Polypeptide S100) whose main components are AMPs on the growth, antibacterial immune and intestinal microbial of Litopenaeus vannamei were study».
Response 1: Thanks for your careful review. As suggested, after communicating with the company, we named the antibacterial component in the fermentation product "WeiGuangSu" to facilitate subsequent purchases by others for repeated tests and other experiments. At the same time, we marked and named it at an appropriate position in the article(Line 26, 114 and 467).
Comments 2: 2) The main changes made by the authors to the manuscript are the replacement of the abbreviation "Antibacterial Fermentation Products" (AFP) with "Fermentation Products" (FP) throughout the text. As well as a not very successful grammatical construction in the introduction.
Response 2: Thanks, the change from "Antibacterial Fermentation Products" (AFP) to "Fermentation Products" (FP) was based on the previous reviewers' suggestion that we did not provide sufficient information on the specific ingredients (peptides) of the product. Therefore, we evaluate it as a whole. As suggested, We made key modifications to the grammar of the introduction (Line 58-105).
Comments 3: 3) Based on the phrase from the Authors reply to Reviewer1: «Readers can screen antimicrobial peptides through the evaluation system established in this study.», I suggest changing the concept of the article from a research-based one to a methodological one. The authors could emphasize that they propose a methodology for assessing the suitability of peptide supplements for fish diets. The result of this work would not be a conclusion that these peptides are good and can be fed to fish, but rather an assessment of how best to evaluate fish health and compare the effectiveness of different health assessment methods. In this case, the focus would shift from the substances themselves to the methods used to assess them.
Response 3: Thanks for your careful review, and we much appreciate your constructive suggestions. It might be that the explanation of "Readers can screen antimicrobial peptides through the evaluation system established in this study" mentioned in our reply is not accurate enough. The core result of this study is that the fermentation products of Bacillus subtilis significantly improved the health status of fish. While obtaining this important result, a series of evaluation indicators set in this study, such as Gut Microbiota, can be used as references for readers when conducting similar evaluations in the future. However, this reference effect is an incidental result of this study and not the core objective. According to the suggestions, we found that among this series of indicators, Gut Microbiota more sensitively reflected that the addition of fermentation products could effectively improve the health status of fish. Meanwhile, we supplemented this conclusion in the summary.
Reviewer 2 Report (Previous Reviewer 2)
Comments and Suggestions for Authors
Reviewers' Comments to Authors:
The manuscript entitled “Fermentation products originated from Bacillus subtilis promotes the hepatic-intestinal health in largemouth bass, Micropterus salmoides” by Liu et al. demonstrates the scientific attempts to investigate the efficacy of fermentation products (FP), which are treated by B. subtilis, supplementation feed in enhancing immune responses, health status, and gut microbiota in an economically important teleost fish under laboratory conditions.
Based on scientific considerations, the manuscript presents interesting findings that contribute to health status, immunity, and gut microbiota in fish induced by FP supplementation feed, with highly possible to imply in fish aquaculture. The content, structure, and overall rationale, and structure are satisfactory with scientific sound. However, it does contain some minor errors and unclear points that the authors must pay more attention to address and improve the quality of this manuscript. The following minor concerns have been left for the authors to improve the quality of the current research work.
Title
Change “promotes” to “promote”.
Materials and Methods
2.2. Antimicrobial Peptide Information
1) Line 124-125. Unitalicize “and”, “(GenBank: GU563992.1)”, “(GenBank: MK034473.1) ” and “(GenBank: OQ283658.1)”.
2.5. Serological Indicator
It was unclear about the sources of serum and the protocol used in this part. Please clarify.
2.7. RNA Extraction and Quantitative PCR Analysis
1) Line 193. Correct “β-Actin”.
2) Line 194. The 2-ΔΔCT method needs its reference.
- Discussion
1) Since the FP from B. subtilis was applied in fish feed, the authors should provide some message to describe and suggest the possible contents and benefits regarding all related parameters that the FB from B. subtilis could influence. This information will be great content that will usefully increase the flow in this section.
2) Line 412. Correct a grammatical error of “a feed additives”.
References
Please carefully correct inconsistent formats and other errors in all references. There are many errors in this part. Please take a look at the correct pattern in the journal guidelines. Additionally, errors in scientific names, punctuation, abbreviations of journal names, and format are always found. Please carefully check one by one.
Author Response
Reviewer 2
Reviewers' Comments to Authors:
The manuscript entitled “Fermentation products originated from Bacillus subtilis promotes the hepatic-intestinal health in largemouth bass, Micropterus salmoides” by Liu et al. demonstrates the scientific attempts to investigate the efficacy of fermentation products (FP), which are treated by B. subtilis, supplementation feed in enhancing immune responses, health status, and gut microbiota in an economically important teleost fish under laboratory conditions.
Based on scientific considerations, the manuscript presents interesting findings that contribute to health status, immunity, and gut microbiota in fish induced by FP supplementation feed, with highly possible to imply in fish aquaculture. The content, structure, and overall rationale, and structure are satisfactory with scientific sound. However, it does contain some minor errors and unclear points that the authors must pay more attention to address and improve the quality of this manuscript. The following minor concerns have been left for the authors to improve the quality of the current research work.
Title
Comments 1: Change “promotes” to “promote”.
Response 1: Thanks for your careful review, and we much appreciate your constructive suggestions. As suggested, we done.
Materials and Methods
2.2. Antimicrobial Peptide Information
Comments 2: 1) Line 124-125. Unitalicize “and”, “(GenBank: GU563992.1)”, “(GenBank: MK034473.1) ” and “(GenBank: OQ283658.1)”.
Response 2: Thanks, done.
2.5. Serological Indicator
Comments 3: It was unclear about the sources of serum and the protocol used in this part. Please clarify.
Response 3: Thanks, the protocol had showed in Line 166-169.“Each fish was subjected to tail vein blood collection after anesthesia, and the collected blood was placed in a 1.5 mL centrifuge tube and left to stand overnight at 4 replications. After centrifugation at 2500 rpm for 15 minutes, the supernatant was taken for serological index detection. ”
2.7. RNA Extraction and Quantitative PCR Analysis
Comments 4: 1) Line 193. Correct “β-Actin”.
Response 4: Thanks, done.
Comments 5: 2) Line 194. The 2-ΔΔCT method needs its reference.
Response 5: As suggested, we added the reference, “Liu, S.; Wang, S.; Cai, Y.; Li, E.; Ren, Z.; Wu, Y.; Guo, W.; Sun, Y.; Zhou, Y. Beneficial effects of a host gut-derived probiotic, Bacillus pumilus, on the growth, non-specific immune response and disease resistance of juvenile golden pompano, Trachinotus ovatus. Aquaculture, 2020, 514, 734446.”
Discussion
Comments 6: 1) Since the FP from B. subtilis was applied in fish feed, the authors should provide some message to describe and suggest the possible contents and benefits regarding all related parameters that the FB from B. subtilis could influence. This information will be great content that will usefully increase the flow in this section.
Response 6: Thank you very much for your valuable opinions. We will discuss the fermentation products as a whole. Further exploration of the impact of their components on the growth and immunity of farmed fish will be our main research content in the future. After communicating with the company, we named the antibacterial component in the fermentation product "WeiGuangSu" to facilitate subsequent purchases by others for repeated tests and other experiments.
Comments 7: 2) Line 412. Correct a grammatical error of “a feed additives”.
Response 7: Thanks, done.
References
Comments 8: Please carefully correct inconsistent formats and other errors in all references. There are many errors in this part. Please take a look at the correct pattern in the journal guidelines. Additionally, errors in scientific names, punctuation, abbreviations of journal names, and format are always found. Please carefully check one by one.
Response 8: Thanks, we have carefully checked through the manuscript.
Round 2
Reviewer 1 Report (Previous Reviewer 1)
Comments and Suggestions for Authors
I've read the authors' responses and reviewed the changes they've made. My main objections seemed to be the unknown composition of the peptide preparation. In the current version of the manuscript, the authors presented the name of this drug with the name of the manufacturer. In this case, I can feel satisfied.
This manuscript is a resubmission of an earlier submission. The following is a list of the peer review reports and author responses from that submission.
Round 1
Reviewer 1 Report
Comments and Suggestions for Authors
This study focuses on an important topic: maintaining the health and commercial viability of largemouth bass in aquaculture. The researchers have carried out and analyzed data on the biochemical effects of dietary supplements on fish, as well as the composition of their gut microbiome and expression of immune genes. Although at first glance, this work seems to have been done in good faith, there is a fundamental flaw in it that essentially renders the data obtained worthless. We do not know which peptide (or peptides, singular or plural, as used in the text) we are referring to.
To begin with, we need more information about the strain of Bacillus subtilis: is it a strain that has been deposited in an international collection of microorganisms, where at least its taxonomic classification has been confirmed?
Further, as is known, Bacillus subtilis strains are able to produce a wide variety of secondary metabolites with antibacterial properties, and not all of these can be classified as peptides (for example, difficidin and bacillaene).
The variety of peptide metabolites produced by bacilli is significant. Which specific metabolites are we discussing in this instance?
How were the antimicrobial peptides produced by this strain identified and developed? How were they purified?
It is difficult to understand from Section 2.2 ....
Thus, I question the validity of conclusions regarding the effect of an unspecified substance or combination of substances on animals, even if they showed an antagonistic effect in vitro. If the authors could provide information on which peptides were used and how they were isolated, purified, or identified, if those are experimentally derived substances, then the work would have practical and scientific value. As it stands, I do not recommend publication.
Next, I drew attention to several minor drawbacks.
L.121 "...and cation"
you varied not only the cations, but also their concentrations.
L.122-123,
What devices did you use to measure the inhibition zones with an accuracy of hundredths of a millimeter?
(See supplementary Table 2). What is the purpose of such precision?
L.147
"finally"
Table 1.
Why are the results presented in thousandths?
L.216
Is it known which particular antimicrobial peptide we are discussing?
L.225
Peptides are already mentioned here in the plural
Figure 5
Please specify the current names of bacterial taxa according to LPSN
https://lpsn.dsmz.de/
L.309
Misprint. It should be "among"
Author Response
Reviewer 1
Comments and Suggestions for Authors
Comments 1: This study focuses on an important topic: maintaining the health and commercial viability of largemouth bass in aquaculture. The researchers have carried out and analyzed data on the biochemical effects of dietary supplements on fish, as well as the composition of their gut microbiome and expression of immune genes. Although at first glance, this work seems to have been done in good faith, there is a fundamental flaw in it that essentially renders the data obtained worthless. We do not know which peptide (or peptides, singular or plural, as used in the text) we are referring to. To begin with, we need more information about the strain of Bacillus subtilis: is it a strain that has been deposited in an international collection of microorganisms, where at least its taxonomic classification has been confirmed?
Response 1: Thanks for your careful review, and we much appreciate your constructive suggestions. As suggested, we provided the information on the Bacillus subtilis strains and antimicrobial peptide components tested and identified by a professional third-party organization, the strain shared approximately 99.86% with Bacillus subtilis DSM 10, Bacillus subtilis NBRC 13719 and Bacillus subtilis JCM 1465, and we also added the key information in the revised manuscript (Line 118-121).
Comments 2: Further, as is known, Bacillus subtilis strains are able to produce a wide variety of secondary metabolites with antibacterial properties, and not all of these can be classified as peptides (for example, difficidin and bacillaene). The variety of peptide metabolites produced by bacilli is significant. Which specific metabolites are we discussing in this instance? How were the antimicrobial peptides produced by this strain identified and developed? How were they purified? It is difficult to understand from Section 2.2 .... Thus, I question the validity of conclusions regarding the effect of an unspecified substance or combination of substances on animals, even if they showed an antagonistic effect in vitro. If the authors could provide information on which peptides were used and how they were isolated, purified, or identified, if those are experimentally derived substances, then the work would have practical and scientific value. As it stands, I do not recommend publication.
Response 2: Thanks for your query. As suggested, the information of antimicrobial peptide components was provided, but the information of how they were isolated and purified could not be afford. Since antimicrobial peptide is the company's product and the related production process belongs to the company's product confidentiality, we cannot get this part of information from the company, which is very regrettable. Currently, we have identified three proteins of the product and compared them with their respective ID numbers, so we can only evaluate the product as a compound additive at present. However, in the future, we will try to isolate and purify the three proteins identified so far, which will serve as the basis for in-depth study of the influence mechanism.
Next, I drew attention to several minor drawbacks.
Comments 3: L.121 "...and cation" you varied not only the cations, but also their concentrations.
Response 3: Thanks, as suggested, we have revised “NaCl, MgCl2, CaCl2 and KCl” to “Na+, Mg2+, Ca2+ and K+”, in Line 129 of the revised manuscript.
Comments 4: L.122-123, What devices did you use to measure the inhibition zones with an accuracy of hundredths of a millimeter?
Response 4: Thanks, we added “by vernier caliper” in Line 130-131.
Comments 5: (See supplementary Table 2). What is the purpose of such precision?
Response 5: In order to prove that antimicrobial peptides can adapt to different temperatures, pH and cationic environments, and have antimicrobial activity.
Comments 6: L.147"finally"
Response 6: Thanks, revised (Line 159).
Comments 7: Table 1. Why are the results presented in thousandths?
Response 7: Usually, the percentile is retained. At first, in order to better retain the SEM value, we chose to retain the thousandth place. After careful consideration and the suggestions of the reviewers, we changed it to the percentile.
Comments 8: L.216 Is it known which particular antimicrobial peptide we are discussing?
Response 8: Thank you for your suggestion, because the product is provided by the company and belongs to the company's trade secret. Currently, we have identified three proteins of the product and compared them with their respective ID numbers, so we can only evaluate the product as a compound additive at present. As for which protein plays the most important mechanism, we are planning to carry out experiments for further study.
Comments 9: L.225 Peptides are already mentioned here in the plural
Response 9: Thanks, we have revised.
Comments 10: Figure 5 Please specify the current names of bacterial taxa according to LPSN https://lpsn.dsmz.de/
Response 10: Thanks, we have revised (Line 234, 237, 240, 245, 256, 285, 287, 292, 295, 321 and 330).
Comments 11: L.309 Misprint. It should be "among"
Response 11: Thanks, done (Line 324).

Reviewer 2 Report
Comments and Suggestions for Authors
Reviewers' Comments to Authors:
The manuscript entitled “Antimicrobial Peptide Derived from Bacillus subtilis Promotes
Micropterus salmoides Liver and Intestine Health” by Liu et al. demonstrates the scientific attempts to investigate the efficacy of antimicrobial peptide (AMPs) supplementation feed in enhancing immune responses, health status, and gut microbiota in an economically important teleost fish under laboratory conditions.
Based on scientific consideration, the manuscript contains interesting findings contributing to immunity and gut microbiota in fish, applicable to fish aquaculture. However, it does contain moderate errors and unclear points that the authors must pay more attention to address and improve the quality of this manuscript to meet an acceptable high-standard journal. The following minor concerns have been left for the authors to improve the quality of the current research work.
Abstract
Line 16. To increase the flow of the manuscript, please add “largemouth bass” before “Micropterus salmoides”.
Line 25. Correct “lipolysis related genes” to “lipolysis-related genes” and the other similar terms throughout.
Line 34. Correct “Anaerobic bacteria” to “anaerobic bacteria” and the other similar terms throughout the manuscript.
- Introduction
Line 83-85. The following sentence's logic is unclear and needs clarification: “These antibacterial metabolites not only enhance the immunity of aquatic animals but also activate the immune system of the organisms through non-specific immune responses [11-13].”.
Line 89-90. The following sentence contains a wrong description: “Antimicrobial peptides have a similar function to antibiotics, regulating the gut microbiota and facilitating the increase in the abundance of beneficial bacteria….”.
Line 90. Correct a grammatical error of relative pronounce “which” and every place throughout the manuscript.
Additional comments
The authors should provide information on the major constraints and the rationale caused by harmful diseases in the target fish, the largemouth bass.
Materials and Methods
2.1. Ethical Approval
Line 110. The stage, city, and country should be added behind “Xinyang Agriculture and Forestry University”.
2.2. Source and Tolerance Tests
1) Please consider revising the topic of this part since it does not represent the correct contents related to the topic.
2) Some key descriptions are poor; these include
2.1) The kind of the AMP from Bacillus subtilis, which is very important to provide energetic information in “Discussion” and any other related part of the manuscript.
2.2) The reference strain of Bacillus subtilis must be declared.
2.3) Sources and preparation of pathogenic bacteria must be clarified based on a good description of basic microbiology.
2.4) The protocols used to conduct each tolerance test are too brief to understand; agar plate, techniques used to prepare each target conditions must be clarified.
2.3. Experimental Diets and Animals
1) The preparation of all experimental feeds is vague. Was top-dressing used for the experimental feed in this section?
2) Clarify “commercial diet”, characteristics, and brand or company of experimental feed should be declared.
3) The following recipes or cookbook style should be avoided; “Suck out the dirt every day and replace one-third of the aquaculture water with fresh tap water that has been aerated for 24 hours.”.
4) Some key water quality parameters, such as DO, ammonia, nitrite, etc., should be properly indicated.
5) One of the most important pieces of information that the authors never clearly declared is the feeding period. The authors always say just a short period, but how long the authors fed experimental feed to fish never described. This information is very serious, similar to the kind of AMP used in the current research.
2.4. Growth Performance Measurements and Sampling
1) Survival rate is not normally classified as a growth parameter, please properly revised.
2) Lines 145-155. Information from “After that, …. detection” should be harmonized and narrated with past tense.
2.5. Serological Indicator
1) This topic must be properly changed since there are biochemical and oxidative stress parameters quantified in the liver and intestine, not in serum. And related information must be corrected throughout the manuscript.
2) If the information in “2.5. Serological Indicator” and “2.6. Hepatic Antioxidant Indices” is the same, they must be merged as one.
2.8. Gut Microbiota Analysis
1) Line 189. Please clarify or refer to the “PCR” protocol.
2.9. Statistical Analysis
1) The statistical parameters to prove “homogeneity and normal distribution” should be referenced or indicated.
2) The following content is not related to “statistical analysis” and must move to merge with “2.7. RNA Extraction and Quantitative PCR Analysis”; “The messenger RNA expression levels of all genes were standardized by β-Actin gene. Gene expression analysis was performed by the 2-ΔΔCT method.”. And please correct “β-Actin gene”.
3) Please revise the following awkward sentence: “Tukey's HSD test and one-way ANOVA were performed to analyze the variation among five treatments in the feeding experimental, and with the P value < 0.05 was set as for significant difference”. Please separately describe “analysis of variance” and “post hoc test” and correctly specify their properties.
4) Please keep consistent in using “p (Italicized)” instead of “P” throughout the manuscript.
Results
Table 1.
1) Correct “Weight gain rate (%)” to “Weight gain (%).”
2) Information on “Total length” and “Average daily gain (ADG)” should be added.
3.2. Serological Indicator
1) Line 235. Based on the previous comment, the term “systemic antioxidant capacity” is wrong since it is locally measured and occurs in the target organs.
2) The following content is wrong described; “the systemic antioxidant capacity (SOD, CAT, GSH-px and T-AOC) of the H1 group was significantly higher than that of the control group, …”. Since GSH-px of H1 is non-significant difference compared to control.
3) The x-axis label should be indicated.
3.3. Hepatic Antioxidant Capacity
1) Line 246-247. Please verify the following content: “the addition of antimicrobial peptides could significantly reduce the MDA content of the liver,”. Since H2 did not significantly differ from the control.
2) To increase the flow of description in Figure 2, Figure 2A-2E should be properly indicated in the text.
3) The x-axis label should be indicated.
3.4. Gene Expression
1) Line 261-262. The terms “H1 and H2” should be added before “could increase…”.
3) The x-axis label of Figures 3A and 3B should be properly indicated.
3.5. Gut Microbiota Diversity and Distribution
1) Table 2. It should be moved and located near section “3.5”.
2) Line 220. Correct “Diversity index…” to “Alpha-diversity indices…”
3) Lines 275-277. The following sentence must be revised since in H1 and H2 groups, Simpson and Shannon indices were not significant differences compared with the control: “…, while the alpha-diversity in the group with added antimicrobial peptides was significantly increased compared to the control group (Table 2, P < 0.05).”.
4) To increase the flow of description in Figure 5, Figure 5A-5C should be properly indicated in the text.
5) Line 289. The following content should be properly modified: “… and Mycoplasma was mainly associated with the H1, H2, and H3 groups.”. Since Mycoplasma was also lightly located in the control group.
6) To increase the flow of description in Figure 6, Figure 6A-6C should be properly indicated in the text.
7) To increase the flow of description in Figure 7, Figures 7A- 7C should be properly indicated in the text.
- Discussion
1) One of the most important contents that the authors should provide energetic discussion in this part is the types of AMP and length of AMP-containing feed application used in this study. The authors should carefully discuss this important and ignorance parts.
2) Line 359. Be careful when describing “acidity and alkalinity”; their meaning are different from “pH”; please revise.
3) Line 363. Please keep consistent in using “Unit/L” throughout the manuscript.
4) Lines 365-366. It’s unfair that the authors discuss that the application of NaCl, MgCl2, CaCl2, and KCl may provide only cations. Since they can release Cl- anions during the experiment and could affect the test AMP. Therefore, Tolerance to ionic strength must be better than cationic concentrations.
5) Lines 404-405. Please revise the following awkward sentence: “As reviewed by Wang et al [49], adding antimicrobial peptides to the diet of farmed animals can improve the immunity of farmed animals,…”.
6) Line 405-406. The following sentence is wrong described: “… and there are reports of reducing fat accumulation, which is consistent with our study.”. Since it was opposite to the current research.
7) Line 435-436. Please verify the following content: “… the infection-induced reduced diversity”.??
8) Line 437. Please verify “ACE”.
References
Please carefully correct inconsistent formats and other errors in all references. There are many errors in this part. Please take a look at the correct pattern in the journal guidelines. Additionally, errors in scientific names, punctuation, abbreviations of journal names, and format are always found.
Comments on the Quality of English Language
See comments
Author Response
Reviewer 2
Reviewers' Comments to Authors:
The manuscript entitled “Antimicrobial Peptide Derived from Bacillus subtilis Promotes Micropterus salmoides Liver and Intestine Health” by Liu et al. demonstrates the scientific attempts to investigate the efficacy of antimicrobial peptide (AMPs) supplementation feed in enhancing immune responses, health status, and gut microbiota in an economically important teleost fish under laboratory conditions. Based on scientific consideration, the manuscript contains interesting findings contributing to immunity and gut microbiota in fish, applicable to fish aquaculture. However, it does contain moderate errors and unclear points that the authors must pay more attention to address and improve the quality of this manuscript to meet an acceptable high-standard journal. The following minor concerns have been left for the authors to improve the quality of the current research work.
Abstract
Comments 1: Line 16. To increase the flow of the manuscript, please add “largemouth bass” before “Micropterus salmoides”.
Response 1: Thank you for your suggestions to improve the manuscript, we have added the “largemouth bass” before “Micropterus salmoides” in Abstract.
Comments 2: Line 25. Correct “lipolysis related genes” to “lipolysis-related genes” and the other similar terms throughout.
Response 2: Done.
Comments 3: Line 34. Correct “Anaerobic bacteria” to “anaerobic bacteria” and the other similar terms throughout the manuscript.
Response 3: Done.
Introduction
Comments 4: Line 83-85. The following sentence's logic is unclear and needs clarification: “These antibacterial metabolites not only enhance the immunity of aquatic animals but also activate the immune system of the organisms through non-specific immune responses [11-13].”
Response 4: Thanks, we have revised this sentence to “These antibacterial metabolites not only enhance the immunity of aquatic animals but also improve the antioxidant capacity of the organisms”
Comments 5: Line 89-90. The following sentence contains a wrong description: “Antimicrobial peptides have a similar function to antibiotics, regulating the gut microbiota and facilitating the increase in the abundance of beneficial bacteria….”.
Response 5: Thanks, we have revised this sentence.
Comments 6: Line 90. Correct a grammatical error of relative pronounce “which” and every place throughout the manuscript.
Response 6: Done.
Comments 7: Additional comments
The authors should provide information on the major constraints and the rationale caused by harmful diseases in the target fish, the largemouth bass.
Response 7: Thanks, we have added information on common diseases of largemouth bass in line 100-101 of the revised manuscript.
Materials and Methods
2.1. Ethical Approval
Comments 8: Line 110. The stage, city, and country should be added behind “Xinyang Agriculture and Forestry University”.
Response 8: Done.
2.2. Source and Tolerance Tests
Comments 9: 1) Please consider revising the topic of this part since it does not represent the correct contents related to the topic.
Response 9: Thanks, the topic maybe make the reader confused, we have changed it to “ Antimicrobial Peptide Information”
Comments 10: 2)Some key descriptions are poor; these include
2.1) The kind of the AMP from Bacillus subtilis, which is very important to provide energetic information in “Discussion” and any other related part of the manuscript.
Response 10: Thank you for your suggestion, because the product is provided by the company and belongs to the company's trade secret. Currently, we have identified three proteins of the product and compared them with their respective ID numbers, so we can only evaluate the product as a compound additive at present. As for which protein plays the most important mechanism, we are planning to carry out experiments for further study.
Comments 11: 2.2) The reference strain of Bacillus subtilis must be declared.
Response 11: As suggested, we provided the information on the Bacillus subtilis strains and antimicrobial peptide components tested and identified by a professional third-party organization, and we also added the key information in the revised manuscript.
Comments 12: 2.3) Sources and preparation of pathogenic bacteria must be clarified based on a good description of basic microbiology.
Response 12: Thanks, we added the information about the pathogenic bacteria.
Comments 13: 2.4) The protocols used to conduct each tolerance test are too brief to understand; agar plate, techniques used to prepare each target conditions must be clarified.
Response 13: As suggested, we have added.
2.3. Experimental Diets and Animals
Comments 14: 1) The preparation of all experimental feeds is vague. Was top-dressing used for the experimental feed in this section?
Response 14: Thanks, the basal diet is the commercial diet (containing 39.64% crude protein, 12.71% crude lipid, 14.01% ash and 10.08% moisture), it does not used the top-dressing.
Comments 15: 2) Clarify “commercial diet”, characteristics, and brand or company of experimental feed should be declared.
Response 15: Thanks, we have added this relevant information.
Comments 16: 3) The following recipes or cookbook style should be avoided; “Suck out the dirt every day and replace one-third of the aquaculture water with fresh tap water that has been aerated for 24 hours.”.
Response 16: Thanks, done. We have deleted.
Comments 17: 4) Some key water quality parameters, such as DO, ammonia, nitrite, etc., should be properly indicated.
Response 17: As suggested, we added these water quality parameters.
Comments 18: 5) One of the most important pieces of information that the authors never clearly declared is the feeding period. The authors always say just a short period, but how long the authors fed experimental feed to fish never described. This information is very serious, similar to the kind of AMP used in the current research.
Response 18: Thanks, we have added the experimental period in line 137.
2.4. Growth Performance Measurements and Sampling
Comments 19: 1) Survival rate is not normally classified as a growth parameter, please properly revised.
Response 19: We have deleted this parameter.
Comments 20: 2) Lines 145-155. Information from “After that, …. detection” should be harmonized and narrated with past tense.
Response 20: Done.
2.5. Serological Indicator
Comments 21: 1) This topic must be properly changed since there are biochemical and oxidative stress parameters quantified in the liver and intestine, not in serum. And related information must be corrected throughout the manuscript.
Response 21: Done.
Comments 22: 2)If the information in “2.5. Serological Indicator” and “2.6. Hepatic Antioxidant Indices” is the same, they must be merged as one.
Response 22: Thanks, the parameter of these two parts is not the same, so we did not merged them.
2.8. Gut Microbiota Analysis
Comments 23: 1) Line 189. Please clarify or refer to the “PCR” protocol.
Response 23: Done.
2.9. Statistical Analysis
Comments 24: 1) The statistical parameters to prove “homogeneity and normal distribution” should be referenced or indicated.
Response 24: Done, line 208.
Comments 25: 2)The following content is not related to “statistical analysis” and must move to merge with “2.7. RNA Extraction and Quantitative PCR Analysis”; “The messenger RNA expression levels of all genes were standardized by β-Actin gene. Gene expression analysis was performed by the 2-ΔΔCT method.”. And please correct “β-Actin gene”.
Response 25: Done.
Comments 26: 3)Please revise the following awkward sentence: “Tukey's HSD test and one-way ANOVA were performed to analyze the variation among five treatments in the feeding experimental, and with the P value < 0.05 was set as for significant difference”. Please separately describe “analysis of variance” and “post hoc test” and correctly specify their properties.
Response 26: Thanks, we have added “Differences among groups were evaluated using one-way analysis of variance (ANOVA). The significance type (linear, quadratic, or cubic) was determined using an orthogonal polynomial comparison. Tukey's HSD test was conducted to assess the significance at a 5% level of confidence with the P value < 0.05 was set as for significant difference.”.
Comments 27: 4)Please keep consistent in using “p (Italicized)” instead of “P” throughout the manuscript.
Response 27: Done.
Results
Table 1.
Comments 28: 1)Correct “Weight gain rate (%)” to “Weight gain (%).”
Response 28: Corrected.
Comments 29: 2)Information on “Total length” and “Average daily gain (ADG)” should be added.
Response 29: Thank you very much for your suggestion. Unfortunately, our experiment period is short, so there is no significant change in growth performance, and there is no statistical body length to calculate, which will be included in the evaluation in future experimental studies.
3.2. Serological Indicator
Comments 30: 1) Line 235. Based on the previous comment, the term “systemic antioxidant capacity” is wrong since it is locally measured and occurs in the target organs.
Response 30: Thanks, we have revised.
Comments 31: 2)The following content is wrong described; “the systemic antioxidant capacity (SOD, CAT, GSH-px and T-AOC) of the H1 group was significantly higher than that of the control group, …”. Since GSH-px of H1 is non-significant difference compared to control.
Response 31: Thanks, deleted.
Comments 32: 3)The x-axis label should be indicated.
Response 32: The x-axis label is name of groups.
3.3. Hepatic Antioxidant Capacity
Comments 33: 1) Line 246-247. Please verify the following content: “the addition of antimicrobial peptides could significantly reduce the MDA content of the liver,”. Since H2 did not significantly differ from the control.
Response 33: Thanks, we rephrased it without “significantly”.
Comments 34: 2) To increase the flow of description in Figure 2, Figure 2A-2E should be properly indicated in the text.
Response 34: Done.
Comments 35: 3) The x-axis label should be indicated.
Response 35: The x-axis label is name of groups.
3.4. Gene Expression
Comments 36: 1) Line 261-262. The terms “H1 and H2” should be added before “could increase…”.
Response 36: Done.
Comments 37: 2) The x-axis label of Figures 3A and 3B should be properly indicated.
Response 37: The x-axis label is name of genes.
3.5. Gut Microbiota Diversity and Distribution
Comments 38: 1) Table 2. It should be moved and located near section “3.5”.
Response 38: Done.
Comments 39: 2) Line 220. Correct “Diversity index…” to “Alpha-diversity indices…”
Response 39: Done. Line 291-292.
Comments 40: 3) Lines 275-277. The following sentence must be revised since in H1 and H2 groups, Simpson and Shannon indices were not significant differences compared with the control: “while the alpha-diversity in the group with added antimicrobial peptides was significantly increased compared to the control group (Table 2, P < 0.05).”.
Response 40: Done,we deleted the “significantly”.
Comments 41: 4) To increase the flow of description in Figure 5, Figure 5A-5C should be properly indicated in the text.
Response 41: Done.
Comments 42: 5) Line 289. The following content should be properly modified: “… and Mycoplasma was mainly associated with the H1, H2, and H3 groups.”. Since Mycoplasma was also lightly located in the control group.
Response 42: Done,we deleted the “ and Mycoplasma was mainly associated with the H1, H2, and H3 groups”.
Comments 43: 6) To increase the flow of description in Figure 6, Figure 6A-6C should be properly indicated in the text.
Response 43: Done.
Comments 44: 7) To increase the flow of description in Figure 7, Figures 7A- 7C should be properly indicated in the text.
Response 44: Done.
Discussion
Comments 45: 1) One of the most important contents that the authors should provide energetic discussion in this part is the types of AMP and length of AMP-containing feed application used in this study. The authors should carefully discuss this important and ignorance parts.
Response 45: Thank you for your suggestion. Because the product is provided by our company, we have identified three proteins and compared them with their respective ID numbers. The rest are commercial secrets. Therefore, at present, we can only evaluate the product as a compound additive. As for which protein plays the most important mechanism, we plan to carry out experiments for further study
Comments 46: 2) Line 359. Be careful when describing “acidity and alkalinity”; their meaning are different from “pH”; please revise.
Response 46: Done.
Comments 47: 3) Line 363. Please keep consistent in using “Unit/L” throughout the manuscript.
Response 47: Done.
Comments 48: 4) Lines 365-366. It’s unfair that the authors discuss that the application of NaCl, MgCl2, CaCl2, and KCl may provide only cations. Since they can release Cl- anions during the experiment and could affect the test AMP. Therefore, Tolerance to ionic strength must be better than cationic concentrations.
Response 48: Thanks, as suggested, we have revised this parts.
Comments 49: 5) Lines 404-405. Please revise the following awkward sentence: “As reviewed by Wang et al [49], adding antimicrobial peptides to the diet of farmed animals can improve the immunity of farmed animals,…”.
Response 49: Done, we changed it to “ As reviewed by Wang et al[49], antimicrobial peptides as a feed additives can improved the immunity of farmed animals and reduced fat accumulation”
Comments 50: 6) Line 405-406. The following sentence is wrong described: “… and there are reports of reducing fat accumulation, which is consistent with our study.”. Since it was opposite to the current research.
Response 50: Revised.
Comments 51: 7) Line 435-436. Please verify the following content: “… the infection-induced reduced diversity”.??
Response 51: We revised the sentence to “Previous studies have reported the sequencing and characterization of hepcidin from the liver of Acrossocheilus fasciatus, and it was found that hepcidin recovered the diversity and compositional in the gut microbiota”
Comments 52: 8) Line 437. Please verify “ACE”.
Response 52: Done.
References
Comments 53: Please carefully correct inconsistent formats and other errors in all references. There are many errors in this part. Please take a look at the correct pattern in the journal guidelines. Additionally, errors in scientific names, punctuation, abbreviations of journal names, and format are always found.
Response 53: Done.

Reviewer 3 Report
Comments and Suggestions for Authors
1. The author is suggested to provide the specific name and version number of the animal husbandry guidelines followed.
2. Author should provide more elaborations on the isolation process of the antibacterial peptide (AMP) from Bacillus subtilis? How was the purity of the AMP verified?
3. Were there any controls used to ensure the accuracy of this concentration 106 CFU/mL author is suggested to write few lines about this?
4.It is not clear from this paragraph about the inhibition zone. How was the diameter of the inhibition zone measured after 24 hours write it?
5. Author is suggested to write How were the percentages of crude protein, crude lipid, ash, and moisture determined?
6. What was the reason for selecting the specific AMP concentrations (1%, 3%, and 5%) for the experimental diets? Were there any preliminary studies or literature references that guided these choices?
7. All the growth parameters are well explained but the author is suggested to add Condition factor also in growth parameters.
8. After how much time were the samples collected from the -80°C storage for analysis?
9. Is there any specific reason for choosing the 2% agarose gel. Author should write a supporting line in this context.
10. While measuring the RNA extraction it was RNA was calculated on nanodrop please mention nanodrop scale on which it was measeured.
11. Author is suggested to write the all genes and their sequence including reference gene in a table. It will be more accurate and good way to present the work.
12. When the genes related stats applied it should be in 2 folds. Author did not write this in statistical analysis. Please write a line
13. This formula is not explained earlier. Author is suggested to write this formula in full explanation above.
14. Results are good but author should add numerical values in the results elaboration.
15. Ethical committee should be added in the last with approval number.

English quality is good
Author Response
Reviewer 3
Comments and Suggestions for Authors
Comments 1: The author is suggested to provide the specific name and version number of the animal husbandry guidelines followed.
Response 1: As suggested, we added the specific name and version number of the animal husbandry guidelines (Line 116)
Comments 2: Author should provide more elaborations on the isolation process of the antibacterial peptide (AMP) from Bacillus subtilis? How was the purity of the AMP verified?
Response 2: Thanks, since we are using the company's product, the purity is labeled as 99.9%, and the separation and purification of the product is a company secret, so we can not provide more information, but we will include the purity in the revision manuscript (Line 118).
Comments 3: Were there any controls used to ensure the accuracy of this concentration 106 CFU/mL author is suggested to write few lines about this?
Response 3: Thanks, we added the methors “The vitality of bacteria in the supplemented diet was evaluated by plate counting on LB medium agar.” (Line 126-127).
Comments 4: It is not clear from this paragraph about the inhibition zone. How was the diameter of the inhibition zone measured after 24 hours write it?
Response 4: Done, we have clarified. “The diameter of inhibition zone was calculated by vernier caliper after 24 h that plate culture under 30℃” (Line 130-131).
Comments 5: Author is suggested to write How were the percentages of crude protein, crude lipid, ash, and moisture determined?
Response 5: Thanks, the data were provided by the company. The whole body moisture and diet moisture were analyzed by drying to a constant mass at 105℃. The Kjeldahl method was used to quantify the crude protein content (Kjeltec 8200, Foss, Denmark). The crude lipid of the crab and the hepatopancreas was extracted with a chloroform/methanol mixture and a 0.37 mol/L potassium chloride solution. Powdered samples were carbonized thoroughly in a carbide furnace (DFD-7000, Lichen, SH, China) and then ashed at 550℃ for at least 6 h in a muffle furnace to quantify the ash content (Line 139-145).
Comments 6: What was the reason for selecting the specific AMP concentrations (1%, 3%, and 5%) for the experimental diets? Were there any preliminary studies or literature references that guided these choices?
Response 6: According to the pre-test of anti-bacterial infection in other fish provided by the company, it is found that 1-5% of the added amount has significant anti-bacterial infection in fish, so we choose 1%, 3% and 5% for the experiment.
Comments 7: All the growth parameters are well explained but the author is suggested to add Condition factor also in growth parameters.
Response 7: Thank you very much for your suggestion. Unfortunately, our experiment period is short, so there is no significant change in growth performance, and there is no statistical body length to calculate the Condition factor index, which will be included in the evaluation in future experimental studies.
Comments 8: After how much time were the samples collected from the -80°C storage for analysis?
Response 8: Thanks, it less than one month.
Comments 9: Is there any specific reason for choosing the 2% agarose gel. Author should write a supporting line in this context.
Response 9: According to preliminary experiments in our laboratory, 2% agarose can best represent RNA banning properties and brightness.
Comments 10: While measuring the RNA extraction it was RNA was calculated on nanodrop please mention nanodrop scale on which it was measeured.
Response 10: Thanks, we had showed it “The quality of total RNA were measured by the 2.0% agarose gel cataphoresis and microspectrophotometer (Thermo, WLM, USA)”, the “ microspectrophotometer (Thermo, WLM, USA)” is nanodrop.
Comments 11: Author is suggested to write the all genes and their sequence including reference gene in a table. It will be more accurate and good way to present the work.
Response 11: Thanks, it showed in Supplementary Table 1.
Comments 12: When the genes related stats applied it should be in 2 folds. Author did not write this in statistical analysis. Please write a line
Response 12: Thanks, we have clarified that “2 folds” in “Gene expression analysis was performed by the 2-ΔΔCT method”, The the 2-ΔΔCT method is “2 folds”.
Comments 13: This formula is not explained earlier. Author is suggested to write this formula in full explanation above.
Response 13: Done.
Comments 14: Results are good but author should add numerical values in the results elaboration.
Response 14: Done (Line 237, 348, 349, 350, 351, 352 and so on).
Comments 15: Ethical committee should be added in the last with approval number.
Response 15: Thanks, we have done.

Round 2
Reviewer 1 Report
Comments and Suggestions for Authors
I see that the authors have worked with the text and made improvements, however, the essence has not changed fundamentally. I still believe that without understanding which peptide we are talking about, work does not make sense. Not only is the production of this peptide a trade secret, readers do not understand what it is about - "some substance" has a positive effect... What is the value of this conclusion? What is the novelty of this work? The authors promise to describe the peptide in detail in the future, but I am not satisfied with this promise...
I believe that it is necessary to identify the product first, to assess whether it is a previously described or a newly discovered substance. If it has already been described earlier, then analyze the literature and then publish a paper showing its positive effects on aquaculture.
In the present form, I do not see a research article, but a scientific report on the testing of a substance, which is appropriate to provide to the customer company, and not to the scientific community.
Author Response
Thanks for your careful review, and we much appreciate your constructive suggestions.The specific information of the antimicrobial peptides can indeed make the experiment more rigorous and complete. Unfortunately, we are unable to provide such specific information as it is a commercial secret .In addition, the antimicrobial peptides have been produced on an industrial scale and has a significant effect on fish health, but its mechanism is still unclear.This study focused on the effects of antimicrobial peptides on serological parameters, hepatic antioxidant Indices, immune-related genes and lipolysis-related genes of liver tissues and gut microbiota of largemouth bass, so as to clarify the reasons for the improvement of fish health caused by antimicrobial peptides.Readers can screen antimicrobial peptides through the evaluation system established in this study. The specific information of antimicrobial peptides does not belong to the focus of this study and will not affect the results of the study.At the same time, we provide the fermentation process of Bacillus subtilis antimicrobial peptides, through this process antimicrobial peptides can be continuously obtained, so as to ensure the repeatability of the results of the effect of antimicrobial peptides on largemouth bass.
Reviewer 2 Report
Comments and Suggestions for Authors
Reviewers' Comments to Authors:
The revised manuscript entitled “Antimicrobial Peptide Derived from Bacillus subtilis Promotes Micropterus salmoidesLiver and Intestine Health” by Liu et al. demonstrates the scientific attempts to investigate the efficacy of antimicrobial peptide (AMPs) supplementation feed in enhancing immune responses, health status, and gut microbiota in an economically important teleost fish under laboratory conditions.
Considerably, the authors have moderately improved most issues suggested previously. However, it still contains someerrors that the authors must pay more attention to address and improve the quality of this manuscript to meet the standards of an acceptable high-standard journal. The following minor concerns have been left for the authors to improve the quality of the current research work.
Simple Summary
Please properly italicize all scientific names throughout.
Materials and Methods
2.2. Antimicrobial Peptide Information
1) Line 132. Unitalicize “and”.
2) The protocols used to conduct each tolerance test are still too brief to understand; the techniques used to prepare each target condition, temperature, pH, and cations, must be clarified.
2.3. Experimental Diets and Animals
1) The preparation of all experimental feeds is vague. Was top-dressing used for the experimental feed in this section?
2) Clarify “commercial diet”, characteristics, and brand or company of experimental feed should be declared.
3) Line 150-156. How is this content involved with information of the current research; “The whole-body moisture and diet moisture were analyzed by drying to a constant mass at 105℃. The Kjeldahl method was used to quantify the crude protein content (Kjeltec 8200, Foss, Denmark). The crude lipid of the crab and the hepatopancreas was extracted with a chloroform/methanol mixture and a 0.37 mol/L potassium chloride solution. Powdered samples were carbonized thoroughly in a carbide furnace (DFD-7000, Lichen, SH, China) and then ashed at 550℃ for at least 6 h in a muffle furnace to quantify the ash content.”. Please carefully check.
2.4. Growth Performance Measurements and Sampling
1) Survival rate is not normally classified as a growth parameter, please revise properly.
2) Lines 145-155. Information from “After that, …. detection” should be harmonized and narrated with past tense.
2.9. Statistical Analysis
1) If the orthogonal polynomial comparison (linear, quadratic, or cubic) was analyzed, the authors should report this information in the manuscript.
2) Line 240. The following wrong and awkward content must be removed, “significance at a 5% level of confidence”.
3) Please keep consistent in using “p (Italicized)” instead of “P” throughout the manuscript.
Results
Table 1.
1) Information on “Total length” and “Average daily gain (ADG)” should be added.
3.2. Serological Indicator
1) Line 271. Based on the previous comment, the term “systemic antioxidant capacity” is wrong since it is locally measured and occurs in the target organs.
2) Figure 1 and the other graph pictures, the x-axis label must be indicated throughout.
3.4. Gene Expression
1) Line 261-262. The terms “H1 and H2” should be added before “could increase…”.
3) The x-axis label of Figures 3A and 3B should be properly indicated.
- Discussion
1) One of the most important contents that the authors should provide energetic discussion in this part is the types of AMP and length of AMP-containing feed application used in this study. The authors should carefully discuss this important and ignorant part.
2) Line 427. Correct a grammatical error of “antibodies, which is…”.
3) Line 476. Correct “Biosynthesis”.
References
Please carefully correct inconsistent formats and other errors in all references. There are many errors in this part. Please take a look at the correct pattern in the journal guidelines. Additionally, errors in scientific names, punctuation, abbreviations of journal names, and format are always found.
Comments on the Quality of English LanguageSee suggestion
Author Response
Reviewer 2
Reviewers' Comments to Authors:
The revised manuscript entitled “Antimicrobial Peptide Derived from Bacillus subtilis Promotes Micropterus salmoidesLiver and Intestine Health” by Liu et al. demonstrates the scientific attempts to investigate the efficacy of antimicrobial peptide (AMPs) supplementation feed in enhancing immune responses, health status, and gut microbiota in an economically important teleost fish under laboratory conditions.
Considerably, the authors have moderately improved most issues suggested previously. However, it still contains someerrors that the authors must pay more attention to address and improve the quality of this manuscript to meet the standards of an acceptable high-standard journal. The following minor concerns have been left for the authors to improve the quality of the current research work.
Comments 1: Simple Summary
Please properly italicize all scientific names throughout.
Response 1: Thanks for your careful review, we have checked throughout the manuscript and revised.
Materials and Methods
2.2. Antimicrobial Peptide Information
Comments 2: Line 132. Unitalicize “and”.
Response 2: Done.
Comments 3: The protocols used to conduct each tolerance test are still too brief to understand; the techniques used to prepare each target condition, temperature, pH, and cations, must be clarified.
Response 3: Thanks, we describe the experimental procedure in detail in lines 127-134. “Prepare ddH2O with pH values ranging from 3 to 11 using 1 mol/L HCl and 5 mol/L NaOH respectively, and dissolve AMP in the ddH2O of different pH values; After dissolving AMP in ddH2O, incubate it in water baths at 20℃, 40℃, 60℃, 80℃, and 100℃ for 1 h respectively; Dissolve AMP in solutions of NaCl, KCl, MgCl2, and CaCl2 with concentrations of 50 mmol/L, 100 mmol/L, 150 mmol/L, and 200 mmol/L respectively. Add 10 μL of AMP with a concentration of 1 mg/mL treated as above into the wells of LB solid plates containing EHEC, and detect the effect of AMP on acid by measuring the diameters of the inhibition zones.”
2.3. Experimental Diets and Animals
Comments 4: The preparation of all experimental feeds is vague. Was top-dressing used for the experimental feed in this section?
Response Response 4: The experimental diets is base on the commercial diet of Guangdong Haid Group Co., Guangzhou, China. We only added the AMP in the diets without top-dressing.
Comments 5: Clarify “commercial diet”, characteristics, and brand or company of experimental feed should be declared.
Response 5: Done. Line 143-144.
Comments 6: Line 150-156. How is this content involved with information of the current research; “The whole-body moisture and diet moisture were analyzed by drying to a constant mass at 105℃. The Kjeldahl method was used to quantify the crude protein content (Kjeltec 8200, Foss, Denmark). The crude lipid of the crab and the hepatopancreas was extracted with a chloroform/methanol mixture and a 0.37 mol/L potassium chloride solution. Powdered samples were carbonized thoroughly in a carbide furnace (DFD-7000, Lichen, SH, China) and then ashed at 550℃ for at least 6 h in a muffle furnace to quantify the ash content.”. Please carefully check.
Response 6: Thanks, we deleted.
2.4. Growth Performance Measurements and Sampling
Comments 7: Survival rate is not normally classified as a growth parameter, please revise properly.
Response 7: Done.
Comments 8: Lines 145-155. Information from “After that, …. detection” should be harmonized and narrated with past tense.
Response 8: Done.
2.9. Statistical Analysis
Comments 9: If the orthogonal polynomial comparison (linear, quadratic, or cubic) was analyzed, the authors should report this information in the manuscript.
Response 9: Thanks, there was a writing mistake and it has been corrected.
Comments 10: Line 240. The following wrong and awkward content must be removed, “significance at a 5% level of confidence”.
Response 10: Done.
Comments 11: Please keep consistent in using “p (Italicized)” instead of “P” throughout the manuscript.
Response 11: Done.
Results
Comments 12: Table 1. Information on “Total length” and “Average daily gain (ADG)” should be added.
Response 12: It is very regretful that since this was a short-term experiment, we did not realize that body length also needed to be counted. However, the ADG is similar to SGR in the results, so we did not added this index.
3.2. Serological Indicator
Comments 13: Line 271. Based on the previous comment, the term “systemic antioxidant capacity” is wrong since it is locally measured and occurs in the target organs.
Response 13: Thanks, we have revised.
Comments 14: Figure 1 and the other graph pictures, the x-axis label must be indicated throughout.
Response 14: Thanks, however, at present, it is more common to label the groups rather than the x-axis label. We have improved the figure caption to avoid ambiguity.
3.4. Gene Expression
Comments 15: Line 261-262. The terms “H1 and H2” should be added before “could increase…”.
Response 15: Done.
Comments 16: The x-axis label of Figures 3A and 3B should be properly indicated.
Response 16: Done.
Discussion
Comments 17: One of the most important contents that the authors should provide energetic discussion in this part is the types of AMP and length of AMP-containing feed application used in this study. The authors should carefully discuss this important and ignorant part.
Response 17: Thanks for your careful review, and we much appreciate your constructive suggestions.The specific information of the antimicrobial peptides can indeed make the experiment more rigorous and complete. Unfortunately, we are unable to provide such specific information as it is a commercial secret. Hence, we couldn’t provide energetic discussion in this part is the types of AMP and length of AMP-containing feed application used in this study.
Comments 18: Line 427. Correct a grammatical error of “antibodies, which is…”.
Response 18: Done.
Comments 19: Line 476. Correct “Biosynthesis”.
Response 19: Done.
Comments 20: References. Please carefully correct inconsistent formats and other errors in all references. There are many errors in this part. Please take a look at the correct pattern in the journal guidelines. Additionally, errors in scientific names, punctuation, abbreviations of journal names, and format are always found.
Response 20: Done.
